# MemLLM: Finetuning LLMs to Use Explicit Read-Write Memory

**Ali Modarressi**[1,2]**, Abdullatif Köksal**[1,2]**, Ayyoob Imani**[1,2]**, Mohsen Fayyaz**[3]**, Hinrich Schütze**[1,2]
*amodaresi@cis.lmu.de*

[1]*Center for Information and Language Processing, LMU Munich, Germany*
[2]*Munich Center for Machine Learning, Germany*
[3]*Microsoft, Berlin, Germany*

**Reviewed on OpenReview:** *https://openreview.net/forum?id=dghM7sOudh*

## Abstract

While current large language models (LLMs) perform well on many knowledge-related tasks, they are limited by relying on their parameters as an implicit storage mechanism. As a result, they struggle with memorizing rare events and with updating their memory as facts change over time. In addition, the uninterpretable nature of parametric memory makes it challenging to prevent hallucination. Model editing and augmenting LLMs with parameters specialized for memory are only partial solutions. In this paper, we introduce MEMLLM, a novel method of enhancing LLMs by integrating a structured and explicit read-and-write memory module. MEMLLM tackles the aforementioned challenges by enabling dynamic interaction with the memory and improving the LLM's capabilities in using stored knowledge. Our experiments indicate that MEMLLM enhances the LLM's performance and interpretability, in language modeling in general and knowledge-intensive tasks in particular. We see MEMLLM as an important step towards making LLMs more grounded and factual through memory augmentation. The project repository is publicly available at: `https://github.com/amodaresi/MemLLM`

## 1 Introduction

State-of-the-art large language models (LLMs) perform well in knowledge-intensive tasks (Yu et al., 2023; Chowdhery et al., 2023). They solve these tasks utilizing the information memorized in their vast array of parameters (Roberts et al., 2020). However, the effectiveness of parameter-based memorization is limited for infrequent entities and concepts (Kandpal et al., 2023; Mallen et al., 2023) and is prone to temporal degradation (Kasai et al., 2023; Jang et al., 2022). Parametric model editing may address some of these issues (Sinitsin et al., 2020; De Cao et al., 2021; Mitchell et al., 2022), but struggles with maintaining locality – possibly damaging model performance in unrelated areas (Yao et al., 2023c; Gu et al., 2024). Moreover, model editing often deteriorates performance when applying sequential editing or batch updates. This is because it primarily focuses on applying (and evaluating) single edits one-by-one (Huang et al., 2023). Finally, model editing may struggle to generalize and maintain previous edits when updating multiple facts simultaneously (Yao et al., 2023c).

Other parametric solutions, like augmenting LLMs with extra parameters such as memory pools can preserve knowledge for subsequent utilization (Wang et al., 2023a; 2024b). However, parametric memorization is prone to distortion and hallucinated nonfactual output. In addition, parametric mechanisms like memory pools have limited capacity and lack interpretability (Maynez et al., 2020; Ji et al., 2023).

Another approach is to augment LLMs with a non-parametric memory component that interacts with the LLM either through natural language text or a formalized API (Wang et al., 2023b). Although prior work has demonstrated enhanced abilities in extended dialogs, long-text generation and question answering

(Packer et al., 2023; Hu et al., 2023; Zhou et al., 2023), these methods are primarily prompt-dependent and necessitate customization for each specific task and model. They also suffer from the lack of a structured memory. This undermines interpretability and interoperability (Wang et al., 2023b). While Retrieval-Augmented Generation (RAG) methods (Lewis et al., 2020) can provide updated facts, their unstructured storage complicates fact editing. Updating an atomic fact requires modifying all related instances to prevent contradictions when conflicting facts are retrieved together (Shi et al., 2023).

In this paper, we introduce MEMLLM, an LLM endowed with an explicit memory component. This architecture has the following key characteristics.

- This explicit memory component has the general advantages of some of the memory-focused work we discussed: we can **keep information accessible indefinitely**, beyond the context window, including **infrequent information** that standard LLMs struggle with.

- The LLM has **both read and write access** to the explicit memory component, i.e., it can store information in the memory as it processes text (or interacts with a user) and retrieve it when it needs it.

- We adopt **finetuning** to train the model to access the explicit memory component through read and write commands. To this end, we specify an **API for read and write access**. Based on the API specification, we create a dataset with training examples of API read and write commands and finetune the LLM on it. Our published training dataset can be used to finetune any language model, endowing it with an explicit memory component without requiring architectural changes.

- The memory component has an explicit structured schema, similar to a database schema. Therefore, it is **interpretable** and inspectable for humans; it is **editable** by humans; it is **scalable** since databases have excellent scalability properties; and it is **interoperable** since the contents of the memory can be exported (e.g., to a different LLM supporting explicit memory) and contents can be imported from data resources (e.g., from Wikidata).

Our evaluation on Re-DocRED (Tan et al., 2022) demonstrates that MEMLLM achieves better perplexity compared to baselines without memory components, with strong gains on named entities. We also show that MEMLLM outperforms non-memory-based methods on knowledge editing.

## 2 Related work

**External memory augmentation.** Augmenting an LLM with memory as an external component can enhance its ability to process larger contexts and maintain reliable records by storing facts and knowledge. Such components include databases, knowledge bases and knowledge graphs that LLMs interact with via natural or formal language (Guu et al., 2020; Lewis et al., 2020; Liu et al., 2022; Yao et al., 2023a; Park et al., 2023; Zhou et al., 2023; Schick et al., 2023; Hu et al., 2023). For instance, Retrieval Augmented Generation (RAG) retrieves relevant text snippets from large document databases, to improve factuality (Guu et al., 2020; Lewis et al., 2020). Other recent solutions store summarized information from previous contexts for future retrieval, improving performance in long-form generation, summarization, question answering and dialog coherence (Park et al., 2023; Zhou et al., 2023; Packer et al., 2023; Chen et al., 2023; Liang et al., 2023).

In general, our framework aligns with external memory methodologies but stands out with its structured format for storing information. This explicit memory facilitates large-scale knowledge editing and makes the model's output generation process more interpretable. While similar structured storage approaches exist, they are often task-specific. For example, Hu et al. (2023) introduce ChatDB, which takes a database structure as input for a data record management task. This requires prior knowledge of the task-specific database schema (e.g., a sales table), which must be defined and provided as a prompt to the model. In contrast, our method is designed for generic language modeling, making it adaptable to a variety of tasks without extensive prompt engineering. Knowledge graphs are another structured format that can be interpreted as a memory of triples, similar to our memory. However, while there is work on extracting triples

from text into knowledge graphs (Zhang & Soh, 2024) using LLMs and on LLMs "interactively" querying knowledge graphs to answer questions (Baek et al., 2023), our approach is distinguished by teaching the LLM memory-write and memory-read functionality through finetuning. This makes memory an integral component of the model's text processing and at the same time enables the model to flexibly handle new and updated knowledge. Similar integrated systems have also been considered in the knowledge graph literature (e.g., AutoKG (Zhu et al., 2024)), but to the best of our knowledge, our system is the first that goes beyond a conceptual proposal and demonstrates empirical success in both language modeling and knowledge editing. Additionally, our published training dataset can be used to endow any trainable language model with explicit memory without requiring architectural changes.

**Memory as a state.** The term *memory* can refer to recurrent architectures that represent past context with vectors (Hochreiter & Schmidhuber, 1997; Cho et al., 2014). Transformer-based models use similar mechanisms with memory tokens (to transfer context across segments) and memory pools (to share information across multiple contexts) (Burtsev et al., 2020; Bulatov et al., 2022; Wang et al., 2024b). While recent advances use vector- or parameter-based memory systems for long-range dependencies (Martins et al., 2022; Wu et al., 2022a;b; Cheng et al., 2023; Wang et al., 2023a; He et al., 2024), they are limited by memory vector capacity (Jelassi et al., 2024). In contrast, MEMLLM has no such architectural limitations and features explicit, interpretable and editable memory.

**Knowledge editing.** The goal of knowledge editing is to apply data-efficient changes to a model's behavior for a set of edits while keeping other knowledge unaffected (Yao et al., 2023c; Gu et al., 2024; Zhang et al., 2024). Meta-learning and locate-then-edit are two classes of parametric methods that modify model weights. In meta-learning, a hypernetwork is trained and applied to the model weights during test time (De Cao et al., 2021; Mitchell et al., 2021). In locate-then-edit, the weights triggered by a knowledge expression are located and modified (Dai et al., 2022; Meng et al., 2022). There are also memory-based methods that do not alter the original model weights but use an external memory (Gu et al., 2024). E.g., methods like SERAC, GRACE and DEFER use retrieval-based memory to fetch previously given edits and apply them to new inputs (Mitchell et al., 2022; Hartvigsen et al., 2024). In WISE (Wang et al., 2024a), in addition to the LLM, two additional parametric models are trained: a side memory and a routing network. Based on the query, the routing network decides which memory to use, the side memory or the main LLM. Evaluations show that multiple edits at a time (batch editing) or successive edits (sequential editing) are challenging tasks – but certainly critical for the intended application of knowledge editing. While most methods can handle a few edits at a time, their performance drops when applying more (Yao et al., 2023c; Wang et al., 2024a). Due to the explicit memory structure of MEMLLM, it can handle a large number of edits while maintaining performance.

## 3 Methodology

Our approach to endowing an LLM with an explicit memory is finetuning with the standard language modeling objective. We now present a finetuning regime that teaches the LLM (1) to extract knowledge from text and write it to the memory and (2) to read knowledge from the memory and leverage it for better language modeling. Following Schick et al. (2023) and Modarressi et al. (2023), we define an API through which the LLM initiates memory writes and memory reads.

### 3.1 Memory structure

The memory stores information in relation triples. Each triple has the form $r = \langle e_s, t, e_o \rangle$, where $e_s$ is the first entity or subject, $e_o$ the second entity or object, and $t$ the relation. Example: $\langle \text{Washington D.C.}, \text{capital of}, \text{United States} \rangle$. The entities and relations are stored as raw text and vectors, each in separate tables. As shown in Figure 1, the facts are stored in the main table "Triple Memory" using three identifiers linked to two other tables: one for subject and object entities, and one for relations. The entities and relations tables are indexed by unique names. We enforce uniqueness across the three linked identifiers in the main table "Triple Memory", i.e., a specific combination (Entity_ID$_1$,Relation_ID,Entity_ID$_2$) can only occur once in the table. Because these identifiers indirectly refer to the names of the entities or relations, different identifiers imply different entity or relation names, thereby preventing redundant storage

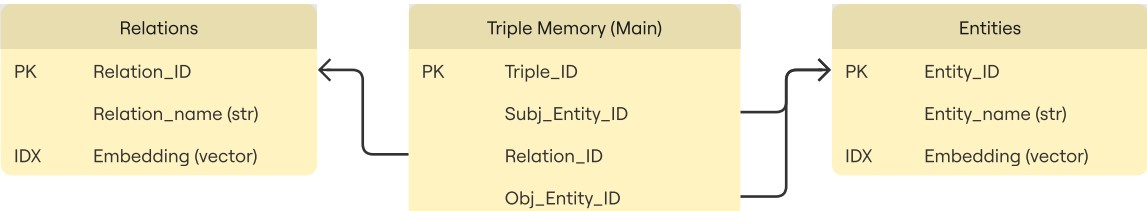

Figure 1: MEMLLM memory schema. Each triple is stored in the "Triple Memory" using its subject entity ID, relation ID and object entity ID, along with a designated triple ID. Entities and relations are stored in separate tables, each containing their designated ID, name and corresponding vector embedding. Both tables are indexed based on their vector embeddings.

of identical facts. The vector representations (created with Contriever (Izacard et al., 2022)) abstract away from different surface forms of the same entity, e.g., "US" vs "USA". In the interest of brevity, we use the symbols $e$ and $t$ for both the entity/relation itself and for its vector.

Our **query format** for querying the memory is:

$$\mathbf{q} \in \{\langle e_s^q, t^q, * \rangle, \langle *, t^q, e_o^q \rangle\}$$

where $e_s^q$, $e_o^q$, $t^q$ are subject entity, object entity, and relation. $*$ indicates the position in the triple of the entity we are querying for. These two templates give us sufficiently specific queries (as opposed to, e.g., queries with two variables) that are likely to return useful entity information.

We want to retrieve triples from the memory that match the query. Given that the surface form of an entity (and also the relation) can vary (e.g., "US" vs "USA"), our match criterion is not exact match, but rather vector similarity. We refer to entities/relations that are similar to the query entity and the query relation as *candidate* entities/relations.

For retrieval, we first determine a set of candidate entities:

$$\mathcal{C} = \{e' | \cos(e^q, e') \geq \tau_e\}$$

That is, all entities with an above-threshold similarity are considered candidate entities.

Similarly, we determine a set of candidate relations:

$$\mathcal{T} = \{t' | \cos(t^q, t') \geq \tau_t\}$$

If the query is a query for an object, i.e., $\mathbf{q} = \langle e_s^q, t^q, * \rangle$, then we retrieve the following final set $\mathcal{E}$ of entities from the memory:

$$\{e_o | \exists (e, t, e_o) \in \mathcal{M} : e \in \mathcal{C}, t \in \mathcal{T}, 0.5(\cos(e, e_s^q) + \cos(t, t^q)) \geq \tau_r\}$$

where $\mathcal{M}$ is the memory. That is, we retrieve all triples with entities/relations from the candidate sets such that their average similarity to query subject and relation is above the threshold. Subject queries are handled analogously.[1]

## 3.2 Memory-API and Inference

We now describe the API that specifies how the LLM initiates memory writes and memory reads.

**Memory writes.** We process the input sentences one by one. See the example given in Figure 2a. For sentence $s_i$ the input $x_i^{\mathrm{MW}}$ to the LLM is formatted as follows:

$$x_i^{\mathrm{MW}} = S_{<i} + (\texttt{\{USER\_ST\}}) + s_i + (\texttt{\{USER\_END\}})$$

where $S_{<i}$ are the $i-1$ preceding sentences and $s_i$ is bracketed by tags to mark it as the *focus sentence*. The LLM's task is then to extract all relations occurring in the focus sentence and to generate a write command

---

[1]We discuss how we set the thresholds and other hyperparameters in Appendix D.

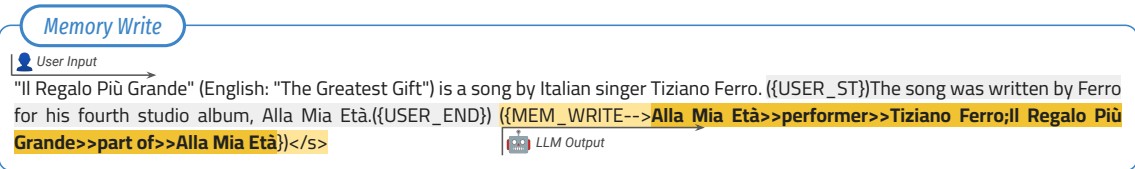

(a) For memory writes, the input is given in two parts. (i) The pretext provides context for the model (e.g., antecedents for pronouns). (ii) The focus sentence is the span of text (bracketed by ({USER_ST}) and ({USER_END})) from which the model is tasked to extract all relations. The model calls the API starting with the ({MEM_WRITE--> command followed by the extracted relations. }) closes the API call. In each document, MEMLLM scans the sentences one by one.

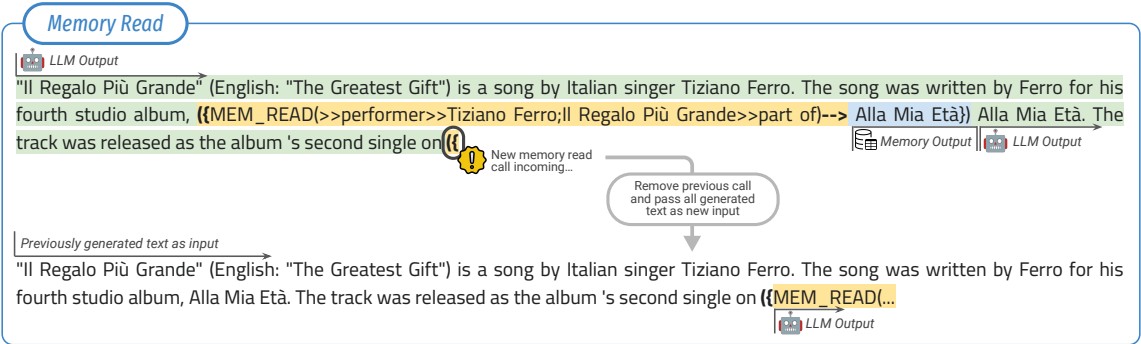

(b) The model decodes one token at a time, as in standard language modeling. It is also trained to generate memory read commands at points when they can retrieve useful information. In the example, after decoding some tokens, the model generates a ({MEM_READ( command followed by queries. --> triggers execution of the queries. Returned results are appended. The model then uses the retrieved results for decoding the posttext. Whenever, during further decoding, the model initiates a new memory read by emitting ({, we remove the previous one because it is unlikely to still be useful.

Figure 2: MemLLM inference with memory read and memory write

that stores them in the memory:

$$y[x_i^{\mathrm{MW}}] = (\{\texttt{MEM\_WRITE-->}e_s^1 \gg t^1 \gg e_o^1; e_s^2 \gg t^2 \gg e_o^2; \dots \})$$

Context $S_{<i}$ is necessary to extract relations from the focus sentence, e.g., if the focus sentence refers to a previously introduced entity with a pronoun. We finetune the LLM to only extract relations from the focus sentence (not from the preceding context); see §3.3 for details.

To extract all relations from a document and write them to the memory, we iterate over the sentences of a document one by one.

**Memory reads.** The LLM can at each point in time either emit a regular token or initiate an API MEM_READ call by generating:

({MEM_READ(

It then continues by generating subject or object queries as introduced above: $\mathbf{q} \in \{\langle e_s^q, t^q, * \rangle, \langle *, t^q, e_o^q \rangle\}$. The syntax for the memory-read API call is:

({MEM_READ$(e_s^{q1} \gg t^{q1} \gg; \gg t^{q2} \gg e_o^{q2}; \dots)$-->

The entity sets $\mathcal{E}$ are then retrieved from the memory (§3.1), merged and appended to the API call:

({MEM_READ$(e_s^{q1} \gg t^{q1} \gg; \dots)$-->$e_1, e_2, e_3, \dots$})

The LLM then continues decoding. Figure 2b gives an example. The LLM starts generating a sentence that refers to an album by the Italian singer Tiziano Ferro. It has learned that just before naming the album is a good point at which to initiate a memory read. Two queries are generated (including: "What is the song "Il Regalo Più Grande" part of?"). One entity is returned by the memory ("--> Alla Mia Età})") and written

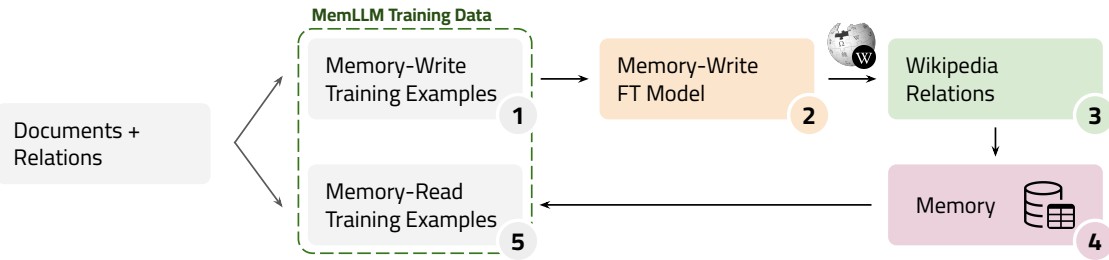

Figure 3: MᴇᴍLLM training data pipeline.

to the buffer. The LLM then generates the name of the correct album ("Alla Mia Età."). This example illustrates that our memory has the potential of reducing hallucinations because through the memory an explicit representation is available of the fact that "Il Regalo Più Grande" is part of the album "Alla Mia Età".

We remove memory-read API calls from the context if they are no longer useful. This happens in three cases: (i) The returned set $\mathcal{E}$ of entities is empty. (ii) The number of retrieved entities exceeds a threshold $Q_{thr}$ ($Q_{thr} = 30$). Such large retrieval results are unlikely to be helpful. (iii) The model emits "({", initiating a new memory-read API call.

For (iii), our motivation for removing the API call is as follows. Omitting API verbiage preserves the text's natural flow and reduces the context to those parts of the input that are still informative for high-quality generation.

### 3.3 Finetuning the LLM

We now describe how we create the dataset for finetuning the model to generate memory-write and memory-read API calls. One innovation of our work is that we create these API training data from corpora annotated with entities and relations, including Re-DocRED (Tan et al., 2022), a Wikipedia corpus annotated in Wikidata format with named entity mentions and relations (see also §4.1) that we will use as an example below.

**Memory-write data.** Figure 3 shows how we use Re-DocRED's annotated relations. For each sentence $s_i$, we retrieve from Re-DocRED all relation triples such that one entity has a full mention (i.e., not a pronoun) in $s_i$ and the other entity has a full mention either in $s_i$ or in the pretext ($S_{<i}$). The memory-write training example consists of the context $x_i^{\mathrm{MW}}$ and the memory-write command $y[x_i^{\mathrm{MW}}]$; see §3.2 and Figure 2a. Since we want to teach the LLM to generate memory-write API calls, we compute the training loss on $y[x_i^{\mathrm{MW}}]$ only.

The set of relation triples can be empty for a sentence $s_i$. In that case we generate a memory-write command in $y[x_i^{\mathrm{MW}}]$ that contains no relations. This encourages the LLM not to generate spurious relations for such "empty" sentences.

**Memory-read data.** For effective memory reads, the LLM has to learn (i) to identify where to initiate a query to the memory, (ii) to generate queries that retrieve helpful information from the memory and (iii) to make good use of the information that is returned by the memory. We now describe how we generate our training data with all three capabilities in mind.

Given a Re-DocRED document $d$, we generate a different training instance $d'$ for each memory-read API call. To produce $d'$, we scan $d$'s annotated entity mentions from the beginning to the end of the document. For each entity mention $e_{\mathrm{target}}$, we collect all relation triples in which it participates. Such triples are a good basis for memory-read API calls that – when issued before $e_{\mathrm{target}}$ first appears – will help the LLM to correctly generate $e_{\mathrm{target}}$; this is why we refer to $e_{\mathrm{target}}$ as the target entity. We keep only that subset of the triples in which the mention of the other entity $e_q$ that the triple refers to (the query entity) has already occurred. (The LLM will in general not be able to generate a query containing $e_q$ if $e_q$ has not yet

occurred.) We also discard all triples that we previously encountered during our scan. (These are already known at this point, so there is little utility initiating a query for them.) We then generate a query for each remaining triple: either $\langle e_q, t, * \rangle$ ($e_{\text{target}}$ = object) or $\langle *, t, e_q \rangle$ ($e_{\text{target}}$ = subject). The memory-read API call for the queries generated for $e_{\text{target}}$ is placed immediately preceding $e_{\text{target}}$. This will retrieve $e_{\text{target}}$ from the memory in many cases and then make it easy for the LLM to correctly predict $e_{\text{target}}$ at the next position.

Next we retrieve results for the query from the memory (see Figure 1). The memory we use here is the one that is populated from Wikipedia by the trained memory-write model (as described earlier in this section and in Figure 2a). The memory write model misses some relations and incorrectly identifies others, resulting in an imperfect memory. We intentionally use this imperfect memory because it aligns the training data with the ultimate inference conditions.

If the query returns a large number of results from memory (more than $Q_{thr} = 30$), we discard it as unlikely to be helpful. (See Appendix B for details.) An example is the query $\langle *, \text{country}, \text{United States} \rangle$ where Wikidata defines the relation "country" as "sovereign state that this item is in". There are thousands of entities that satisfy this query. Such an unspecific result is not useful. Otherwise we add the queries and the query result to $d'$; see Figure 2b and §3.2.

Finally, we add the rest of $d$ to $d'$ up to the next memory read (indicated by "`({`") or (if there isn't one) the entire remainder of $d$.

To summarize, each training example $d'$ is a concatenation of (i) the pretext, including the first two letters ("`({`") of the API call, (ii) the API call proper "`MEM_READ(`$e_s^{q1}$»$t^{q1}$»`...)-->`", (iii) the query result from the memory "$e_1, e_2, e_3, \ldots$`})`" and (iv) the posttext. The posttext consists of the rest of the following text until the next memory read or (if there isn't one) the entire rest of $d$.

The loss is applied to the API call (ii) – this teaches the model to generate the correct API call. The loss is also applied to the posttext – this teaches the model (a) to make good use of the information provided in the query result for predicting entities and (b) to predict the next memory read (as indicated by "`({`"). (iii) is not subject to the loss since the query results are generated by the memory, not by the LLM. For the training example $d'$ that contains the very first "`({MEM_READ(`" in the document (and only for this $d'$), the loss is also applied to the pretext – because the LLM needs to learn where to generate this first "`({MEM_READ(`". For a more comprehensive overview of the memory-read data generation process, refer to Appendix C, which outlines the detailed algorithm.

# 4 Experiments

## 4.1 Setup

To train and evaluate MEMLLM, we construct training and evaluation datasets as described in §3.3. We require datasets annotated with entities and relations. We use three such datasets. (i) Re-DocRED (Tan et al., 2022): Wikipedia texts annotated (in a Wikidata format) with named entity mentions, coreference information and 96 relations (occurring intra- and inter-sentence). Re-DocRED includes many relation instances missing in DocRED Yao et al. (2019). (ii) DocRED's distant supervised training set. It includes >100K documents but fewer relations per document. The size of this dataset makes the training more effective and robust. (iii) A set of "counterfactual" variations of Re-DocRED (Modarressi et al., 2024). Modarressi et al. (2024) introduces an entity replacement strategy to find and apply suitable replacements over Re-DocRED. In our initial tests, we found that teaching the model to produce counterfactual answers (which often contradict its parametric memory) increases robustness against pretrained knowledge bias. This is described in detail in §4.1.1.

We finetune two Mistral-7B (Jiang et al., 2023) models using LoRA (Hu et al., 2022), a memory-write model and a memory-read model. See Appendix D for details on finetuning and hyperparameters. Our baselines are the original Mistral-7B and the memory-read model with its memory capabilities disabled. The latter baseline lets us ascertain to what extent improvements are due to in-domain finetuning (as opposed to the memory).

| Filtering Approach | Prec. | Rec. | F1 | Acc. |
|---|---|---|---|---|
| Baseline | 0.58 | 0.83 | 0.68 | 0.61 |
| Justification | 0.56 | 0.82 | 0.66 | 0.59 |
| Reasoning | **0.78** | **0.84** | **0.80** | **0.80** |

Table 1: Comparing performance of different prompting strategies for filtering distant supervision data. The reasoning approach similar to chain-of-thought prompting performs best among the strategies.

#### 4.1.1 Filtering Distant Supervision Relations

Re-DocRED is human-annotated and mostly consists of relations with explicit evidence. In contrast, the distant supervised DocRED training set lacks explicit evidence and contains many false positives due to its automated annotation method. To address this, we implement a few-shot-based filtering approach to remove false-positive relations. We also apply this filtering to Re-DocRED relations that lack explicit evidence.

To increase the number of training examples, we also include examples from the distant supervision subset of DocRED. Distant supervision (Mintz et al., 2009) assumes that a relation $r$ exists between two entities $(e_s, e_o)$ in a text if the text includes both entities and the $r = \langle e_s, t, e_o \rangle$ relation triple exists in a knowledge base. While this method is valuable for relation extraction, it may introduce noisy examples without any evidence of the relation in the text. This noise could adversely affect our training pipeline.

The experimental setup is as follows: We start with a partial document ($S = \{s_1, s_2, \ldots, s_i\}$) mentioning two entities ($e_1, e_2$), with at least one of them present in the last sentence (i.e., the focus sentence), $s_i$. Our aim is to determine whether the potential relation $r$ between $e_1$ and $e_2$ has any evidence in the last sentence.

To filter out negative examples, we use large language models (i.e., Mixtral). We design 8-shot in-context learning examples to detect if there is evidence of a relation in the focus sentence. We curate a test set to evaluate the performance of this filtering mechanism as follows. We select 1000 examples from the human-annotated split of DocRED as positive examples where the focus sentence is annotated as evidence. For negative examples, we choose 1000 examples where the focus sentence contains at least one entity but there is no evidence for the relation in the focus sentence.

For prompting, we apply three different strategies. In the first approach (baseline), we expect the LLM to answer with "Yes" or "No" to report whether the focus sentence contains evidence. With the second approach (justification), we expect the LLM to provide justification after giving the answer. In the final approach (reasoning), we expect the LLM to generate a natural sentence representing the relation, then provide reasoning, and finally generate the answer with "Yes" or "No" in the last sentence, similar to chain-of-thought prompting.

We present the results in Table 1. These results suggest that the reasoning approach outperforms the other two approaches by a large margin. Also, it suggests that the filtering would lead to higher quality based on the high recall score, 0.84. We demonstrate the best-performing prompt in Appendix E.

After applying this method, we use the filtered distant dataset alongside 10 counterfactual variations of Re-DocRED to generate data for the initial fine-tuning phase. We then continue the finetuning process with data generated from the supervised set of Re-DocRED.

### 4.2 Perplexity evaluation

To evaluate how the memory component would improve language modeling, we perform a perplexity evaluation. For this, we need a corpus to extract facts, do memory writes, and store them in structured memory. Our primary source is a full dump of English Wikipedia[2], but we also evaluate using Wikipedia abstracts and Re-DocRED texts for further analyses. The language modeling evaluation then examines how the model performs in terms of perplexity once the memory component has been filled. Following Liu et al. (2022), we report: (1) OVERALL PPL (PPL on the entire input text), (2) TARGET PPL (PPL on the target enti-

---

[2]Dump date: 2023-11-01, available at: `https://huggingface.co/datasets/wikimedia/wikipedia`

| | Memory | PPL | | |
| --- | --- | --- | --- | --- |
| | | **OVERALL** | **TARGET** | **ENTITY** |
| Baseline #1 (Mistral-7b) | (no memory) | 5.823 | 3.550 | 4.666 |
| Baseline #2 (Memory Disabled) | | 4.997 | 3.510 | 4.353 |
| ① **MemLLM** | MW[*Wikipedia (Full)*] | 4.905 | 2.986 | 4.187 |
| ② MEMLLM | MW[*Wikipedia (Abs.)*] | 4.898 | 2.955 | 4.170 |
| ③ MEMLLM | MW[*Re-DocRED (Test)*] | 4.863 | 2.821 | 4.102 |
| ④ + Gold MR Pos. & Queries | | 4.634 | 1.938 | 3.548 |
| ⑤ MEMLLM | Re-DocRED (Test) | 4.811 | 2.596 | 3.993 |
| ⑥ + Gold MR Position | | 4.674 | 2.232 | 3.728 |
| ⑦ + Gold Queries | | 4.431 | 1.364 | 3.149 |
| ⑧ + Gold Target | | 4.426 | 1.357 | 3.142 |
| ⑨ + Gold Target Only | | 4.385 | 1.194 | 3.026 |

Table 2: MEMLLM performance on OVERALL PPL (all text), TARGET PPL (target entities) and ENTITY PPL (all entities). We show the effect of memory content ("**Memory**"). "MW[$X$]": the memory is populated with triples generated by memory-writes with MEMLLM run on $X$. ⑤–⑨: the triples are from Re-DocRED (Test), Re-DocRED's validation set.

ties) and (3) ENTITY PPL (PPL on all named entities). The model produces a token $w_i$ with probability $p(w_i|w_{<i})$:

$$p(w_i|w_{<i}) = p(w_i|w_{<i}, \text{MR})p(\text{MR}|w_{<i})$$
$$+ p(w_i|w_{<i})(1 - p(\text{MR}|w_{<i}))$$

where $p(\text{MR}|w_{<i})$ is the probability of initiating a memory read (MR) with the "({" token.[3]

In case of MR, $w_i$ is conditioned on both MR (including the MR call and the returned result, see Figure 2b) and the pretext $w_{<i}$. If there is no MR, then $w_i$ is only conditioned on $w_{<i}$.

Table 2 gives perplexity results on Re-DocRED test. MEMLLM outperforms the two baselines on all three PPL measures (①). This increase for triples appearing for the first time in the text suggests that memory-reads successfully recall relevant information for language modeling. This improvement benefits not just all entities in the text (ENTITY PPL) but the entire text (OVERALL PPL). TARGET PPL (the focus of this work) improves by .524 (2.986 vs 3.510). This substantial improvement demonstrates the effectiveness of MEMLLM for target entities. This capability is crucial for generating factual text and preventing hallucinations.

For our **memory-read analysis,** instead of using the LLM to write to the memory, we directly populate the memory with the relations from the validation set (indicated as "Re-DocRED (Test)" in column "Memory"). This lets us investigate what would happen if the memory-write process were error-free, i.e., all remaining errors are due to the memory-read process. We look at four potential sources of error in memory reads and present in each case an ablation in which this source of error is eliminated: the position of the memory read is the gold position immediately before the target entity (⑥ "Gold MR Position"), the query to the memory is the gold query (⑦ "Gold Queries"), the correct target entity is returned by the memory (⑧ "Gold Target"), the correct target entity is returned by the memory and no other entities (⑨ "Gold Target Only"). ⑨ is the lower bound perplexity for perfect memory reads (and perfect memory writes).

---

[3]We evaluate $p(w_i|w_{<i})$ by setting $p(w_i|w_{<i}, \text{MR})$ to zero for all positions except for positions where memory reads actually occur. The reason is that taking into account an MR call at each position results in a tree with $2^n$ leaves at position $n$ in the text, each requiring a memory call. This is too expensive to compute. As a result, we evaluate with smaller values of $p(w_i|w_{<i})$ than the true $p(w_i|w_{<i})$ estimated by the model and, consequently, with higher perplexities, thus unfairly penalizing MEMLLM. Note that this is a problem for fairness of our perplexity evaluation, but not for a real application (where we only pursue a single path at each point).

| Method | REL | GEN | LOC | AVG |
|--------|-----|-----|-----|-----|
| DEFER | 0.02 | 0.02 | 0.67 | 0.24 |
| GRACE | **1.00** | 0.02 | **1.00** | 0.67 |
| WISE | 0.70 | *0.67* | **1.00** | *0.79* |
| **MemLLM** | *0.78* | **0.76** | *0.97* | **0.84** |

Table 3: Knowledge editing results on ZsRE with 1000 sequential edits. AVG: mean of reliability (REL), generalization (GEN), locality (LOC). Baseline results (using the same model, Mistral-7B, and edit set) are from Wang et al. (2024a). Bold (italics): (second) best result.

Comparing ⑨ and ⑧ on TARGET PPL (1.194 vs 1.357) shows the effect of "ambiguity". The +.343 gap is due to the memory returning more targets than just the gold target.

Moving from ⑧ to ⑦ (1.357 to 1.364) indicates the impact of the memory retrieval process. In ⑦, we use the gold queries, but without ensuring the inclusion of the gold target in the results. The next comparison highlights the impact observed when the LLM itself generates queries (⑥) vs when only gold queries are issued (⑦). Finally, the effect of the model itself selecting the position for the memory read (⑥) versus predetermining that position (⑤) is shown in the rise from 2.232 to 2.596.

To isolate the factor **memory-write performance**, we fix (i) gold memory-read positions and queries and (ii) the input corpus for extracting relations (we use Re-DocRED test). We only vary the method by which the memory is populated: running the memory-write model on the input corpus (④) vs reading out the relations from the gold data and directly storing them in memory (⑦). As expected, PPL improves when directly stored (i.e., 100% precision and recall) triples are used (⑦) vs when MEMLLM extracts and writes triples to memory (④). This indicates that there is room for improvement by training MEMLLM to do a better job at information extraction.

**Scaling the stored triples.** In a real-world scenario, the size of the memory and, consequently, the size of query results will get large. This increases the risk of unhelpful information being returned from the memory. To investigate this, we compare our main experiment (①, using the full Wikipedia, 111M triples) with two ablations that use only Wikipedia abstracts (②, 38M triples) and only Re-DocRED test (③). Table 2 shows that there is a relatively small negative effect of memory size: ③ (memory stripped down to the relations generated from Re-DocRED test) is only slightly better than ① (full memory). This suggests good scaling properties of our approach.

**Memory redundancy reduction benefits.** Instead of storing facts in a structured triple format, each fact could be stored as its own proposition (Chen et al., 2024). For instance, one can store the triple ⟨Washington D.C., capital of, United States⟩ as the proposition: "Washington D.C. is the capital of the United States." Based on the memory populated with full Wikipedia (111M triples), an equivalent RAG system would need to store and index 111M proposition sentences. In a vector-based (dense) retrieval setting, this means storing and indexing 111M vectors alongside their text counterparts. However, in MEMLLM, we store these facts in structured triples with identifiers (cf. Section 3.1), and the only vector-based indices are entity and relation embeddings. After storing the 111M triples, the entity/relation tables contain roughly 21M unique entity/relation records. Thus, as the number of extracted facts increases, our structured

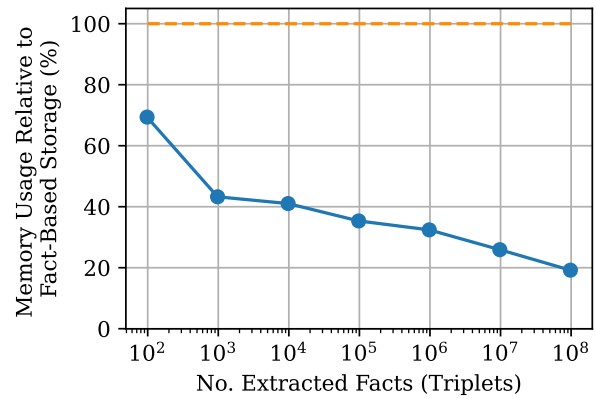

Figure 4: Memory efficiency of storing structured triples vs. proposition-based storage. The y-axis represents the fraction of memory required compared to a RAG system that stores an embedding per fact.

triple approach results in significantly lower mem-
ory usage and, at large scales, it requires less than
20% of the memory (21M) needed for direct proposition storage (111M), demonstrating its efficiency in reducing redundancy (Figure 4). Moreover, having only entities and relations encoded as vectors reduces ambiguity and improves recall by introducing less noise compared to encoding entire sentences.

## 4.3 Knowledge Editing Evaluation

To test whether MemLLM facilitates knowledge editing, we evaluate prompt-based knowledge editing. Following Hartvigsen et al. (2023) and Yao et al. (2023b), we measure reliability (REL), generalization (GEN) and locality (LOC). Each example includes a prompt, an edit, a generalization test prompt and a locality test prompt. The task is to apply the edit on the original prompt to the model. The goal is for the model to respond to original and generalization test prompts in accordance with the edit. The locality test checks whether unrelated knowledge is affected. An ideal method effectively applies edits, generalizes correctly and does not harm unrelated knowledge.

Following Wang et al. (2024a), we evaluate MemLLM on ZsRE, a closed-book question answering dataset (Levy et al., 2017) with locality prompts selected from Natural Questions (NQ) (Kwiatkowski et al., 2019). We apply 1000 edits from the evaluation set by appending them to the end of the questions (the prompts) using the following text: "It is or they are" and bracketing them with tags. Example: "(`{USER_ST}`)What city was Luca Verdecchia born? It is or they are Naples(`{USER_END}`)." The memory-write model should then extract and store Verdecchia's place of birth, i.e., Naples. We evaluate MemLLM using a 5-shot QA prompt. The first four examples are typical question-answer pairs. The fifth in addition includes a full memory-read call. A prompt – a generalization or locality test prompt – is appended to the 5 shots and a memory-read API call executed after the question mark.

We expect the model to answer the questions based on the memory filled with the edits. Some edits in the dataset overlap or are intended to replace previous edits. Therefore, if a newly extracted triple has an exact matching entity and relation with an old triple, we replace the old one with the new one.

Table 3 compares knowledge editing results for MemLLM with three baselines. MemLLM outperforms the baselines (AVG of .84). High reliability (.78) and generalization (.76) scores suggest that MemLLM (i) manages to extract and store the relation triple based on the edit and (ii) utilizes the edit in the memory-read process to answer the original and the generalization test questions correctly. Moreover, since MemLLM uses an explicit memory the applied edits only mildly affect the answers to unrelated questions: MemLLM has a score of .97 on locality. This indicates that there is little cumulative deterioration of the explicit memory.

**Qualitative Analysis.** Leveraging MemLLM's interpretable design, we identified the causes behind the 22% performance gap in reliability from 1.00 to 0.78 (REL, MemLLM vs GRACE). Out of 216 errors, 45 were due to memory writes resulting in no triples or triples without the desired edit. In 95 cases, the edit was captured in the memory write but not retrieved by the memory read, either due to a bad query or incorrect relation extraction during the memory write. Another 63 errors occurred when the model did not effectively use the edits even though they were correctly retrieved. Many of these errors are due to the limitation to 96 relations (see §4). For example, the question "How endangered does the IUCN consider Hyloxalus parcus?" involves a relation that is not covered: "IUCN conservation status". In another case, the question "What family lineage was Xiao Jia part of?" retrieves the correct edit ("Southern Ming Dynasty") but for an incorrect relation: (`{MEM_READ(Xiao Jia»country of citizenship»)}`), as the relation "family" is not one of the covered 96. The model may then not recognize the query result as relevant to the question and ignore it. Addressing this limitation by supporting more relations would resolve many of these errors. Even with this limitation and not being specifically designed for knowledge editing, MemLLM outperforms other model editing methods. We believe that expanding its capability to handle a broader range of relations would greatly enhance its performance.

## 5 Conclusion

We present MEMLLM, a novel approach to endowing an LLM with an explicit structured memory. We publish a training dataset that can be used to extend any standard LLM with such a memory. We show that MEMLLM improves language modeling (as measured by entropy) and outperforms state-of-the-art knowledge editing methods on ZsRE.

## Limitations

While the structured relation-based memory improves factuality and interpretability, it has its own limitations. The current version of MEMLLM handles only 96 relation types commonly used in Wikidata. However, to handle all types of knowledge extraction and storage, the model should be capable of extracting other types of relations. Composite relations that could be inferred from multiple already extracted relations are not detected or utilized in the current version of MEMLLM. For instance, if we extract (California, country, United States) and (Apple Inc., located in, California), we expect the relation (Apple Inc., located in (or Country), United States) to be inferred. MEMLLM is not a memory-aware solution. This means if a fact is not stored in the memory, but the decoding process generates a partial prompt that requires that fact, the model would either continue generation based on its parametric knowledge or hallucinate.

We acknowledge that Retrieval-Augmented Generation (RAG) methods are widely used for a more grounded text generation. However, like other baselines in knowledge editing, we do not include RAG methods in our comparisons due to the difficulty of modifying facts in their unstructured format. Updating a single fact would require locating and altering every related text snippet and embedding within the RAG's knowledge base, which is highly impractical. MemLLM's memory is not pre-populated as a KB; it stores edited facts through a memory-write process, similar to non- and semi-parametric editing methods. While methods like ChatDB (Hu et al., 2023) also employ non-parametric structured memory, they rely on database tables that are more suitable for analytical tasks. This setup requires the user to define a task-specific database schema in advance and explicitly prompt the model with it. In contrast, MemLLM uses a much more generalized memory structure that supports both language modeling and QA tasks (as discussed in the knowledge editing section) without requiring any changes to its approach or memory format. Furthermore, the nature of prompt-based approaches—compared to fine-tuning—makes them inherently less faithful and more dependent on the specific content stored in memory.

We refer all these limitations to future work, as in this paper we have laid the initial groundwork for building a more complex and comprehensive method.

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

## A  Memory-write Decoding Method

While one might use MEMLLM with greedy decoding for memory writes, we suggest that the finetuned model may end the memory-write too early, before completely extracting all relations. Therefore, to ensure the model captures all relevant relations, we implement a late stopping strategy. In this approach, similar to greedy decoding, we consistently select the top-scoring token as the next token, unless it's the closing token "}". If the closing token scores highest, we note its position, calculate the average log probability score of the sequence up to that point, and proceed with the second highest scoring token—-typically the ";" separator—-resuming greedy decoding. By tracking the positions where the closing token was predicted, along with their

| Query ($\mathbf{q}$) | Relation type ($t^q$) |
|---|---|
| $\langle *, t^q, e_o^q \rangle$ | country of citizenship, country, country of origin, religion, place of birth, place of death, work location, location, basin country, residence, location of formation, publication date, production company, platform, original language of work, applies to jurisdiction, located in the administrative territorial entity, headquarters location, inception, employer, date of birth, date of death, educated at |
| $\langle e_s^q, t^q, * \rangle$ | contains administrative territorial entity |

Table 4: List of ambiguous queries subject to the filtering process.

corresponding logprob scores, we maintain the generation process until there are no enhancements in the scores for K=5 consecutive times. Subsequently, we halt the generation and select the position with the highest score as the cutoff point.

## B  Filtering Ambiguous Queries

As we aim to assist the model with the stored memory content, having concise query results would facilitate reaching this objective. Getting precise outputs from the memory would require queries that are tailored in a way which lead to an exact match or related entities to the target entity. To reduce the chances of getting a vast and wide-range amount of outputs from the memory, we exclude queries that potentially leads to such results. In Table 4, we demonstrate query patterns that we intuitively assume based on the queried entity and the relation type that would lead to an ambiguous result. Therefore, we drop any query that would match with one of the mentioned patterns.

## C  Memory-read Data Generation

Algorithm 1 presents the pseudocode for the process of generating MEMLLM 's memory-read training data. See Section 3.3 for a detailed description of the same process.

## D  Hyperparameters Details

We finetune MEMLLM, with a Mistral-7B-v0.1 model (Jiang et al., 2023) using an Adam optimizer (Kingma & Ba, 2015), with the learning rate set to $2 \times 10^{-5}$, 2 epochs, and a batch size of 96. For LoRA specific parameters, we apply a dropout rate of 0.1, with a rank of 16 and an alpha weight of 8.

We opted to set $Q_{thr}$ to 30 based on the distribution of triples observed in the Re-DocRED dataset. During the construction of memory-write data, we found that the 95th percentile of sentences contained a maximum of approximately 30 triples. This value serves as an upper threshold: if the number of entities exceeds 30, it is likely to surpass the typical number of triples within a sentence. Consequently, a higher count of entities could indicate that many of them are unrelated to the sentence's factual content, reducing their overall informativeness.

To select the memory retrieval hyperparameters (§3.1), we must balance explicitness with the need to accommodate variations in entity mentions and relation types. This balance is influenced by the data and the entities involved, but generally, a larger $\tau_e$ increases explicitness. However, it also limits the number of similarly mentioned entities that can be retrieved, which depends on the use case. A smaller $\tau_e$ could retrieve more entities, but it would also increase query execution time. The selection of $\tau_t$ depends on the supported relation types and the required flexibility in retrieving closely related relation types. For instance, in model editing, where handling loosely similar relation types is necessary, a more relaxed $\tau_t$ value is appropriate.

---

**Algorithm 1** Memory-read Data Generation

---

**Input:**
   $D$: Re-DocRED documents ($d \in D$)
**Output:**
   $\mathcal{D}_{\mathrm{MR}}$: Memory-read training data ($d' \in \mathcal{D}$)
**Auxiliary functions:**
   TRIPLES($e_{\mathrm{target}}, d$): Returns all triples in document $d$ containing the entity $e_{\mathrm{target}}$ as the subject or object:

$$\mathrm{TRIPLES}(e_{\mathrm{target}}, d) = \{\langle e_{\mathrm{target}}, t, e_q \rangle \in d\} \cup \{\langle e_q, t, e_{\mathrm{target}} \rangle \in d\}$$

   QUERYRESULT($q, M$): Returns the set of entities retrieved by query $q$ over memory $M$ (e.g. Wikipedia facts extracted using memory writes).
**Definitions:**
   positionIdx: The position of an entity (or word) within the document text.
1: Initialize SeenTriples ← {}, SeenEntities ← {}, $\mathcal{D}_{\mathrm{MR}}$ ← [ ]
2: **for** $d$ in $D$ **do**
3:     prevReadPos ← 0, $\mathcal{D}_d$ ← [ ]
4:     **for** $e_{\mathrm{target}}$ in $d$, in order of appearance in text **do**
5:         $\mathbf{q}$ ← {}, $\mathbf{R}$ ← [ ]                            ▷ ($\mathbf{R}$ is the query-results dictionary for the queries in $\mathbf{q}$)
6:         **for** triple in TRIPLES($e_{\mathrm{target}}, d$) **do**
7:             **if** triple ∉ SeenTriples **then**
8:                 Let $e_q$ be the other entity in triple
9:                 **if** $e_q \in$ SeenEntities **then**
10:                     $q$ ← triple − $e_{\mathrm{target}}$
11:                     **if** $q \notin \mathbf{q}$ **then**
12:                         $R_q$ ← QUERYRESULT($q, M$)
13:                         **if** $|R_q| \leq Q_{\mathrm{thr}}$ **then**
14:                             Add $q$ to $\mathbf{q}$, $\mathbf{R}[q]$ ← $R_q$
15:                         **end if**
16:                     **end if**
17:                 **end if**
18:                 Add triple in SeenTriples
19:             **end if**
20:         **end for**
21:         **if** $|\mathbf{q}| > 0$ **then**
22:             currentReadPos ← $e_{\mathrm{target}}$.positionIdx
23:             $d'$.pretext ← $d$.text[:currentReadPos]
24:             $d'$.queries ← {}
25:             $d'$.results ← {}
26:             Sort $\mathbf{q}$ in ascending order by $|\mathbf{R}[q]|$
27:             **for** $q$ in the first 3 elements of $\mathbf{q}$ **do**
28:                 Add $q$ to $d'$.queries
29:                 $d'$.results ← $d'$.results ∪ $\mathbf{R}[q]$
30:             **end for**
31:             **if** $|d'$.results$| = 0$ **then**
32:                 $d'$.results ← {$e_{\mathrm{target}}$}
33:             **end if**
34:             **if** $|\mathcal{D}_d| > 0$ **then**
35:                 $\mathcal{D}_d[-1]$.posttext ← $d$.text[prevReadPos:currentReadPos]
36:                 prevReadPos ← currentReadPos
37:             **end if**
38:         **end if**
39:         Add $e_{\mathrm{target}}$ in SeenEntities
40:     **end for**
41:     $\mathcal{D}_d[-1]$.posttext ← $d$.text[currentReadPos :]
42:     $\mathcal{D}_{\mathrm{MR}}$ ← $\mathcal{D}_{\mathrm{MR}} \, \| \, \mathcal{D}_d$                            ▷ ($\|$ denotes list concatenation)
43: **end for**
44: **return** $\mathcal{D}_{\mathrm{MR}}$

---

Finally, $\tau_r$ determines the final number of outputs retrieved during the memory-read. A larger $\tau_r$ makes the memory more explicit in both entity and relation type. We set $\tau_e$ and $\tau_t$ to 0.7 and $\tau_r$ to 0.85. We set these values to $\tau_e = 0.85$, $\tau_t = 0.2$ and $\tau_r = 0.6$ respectively for model editing experiments.

# E    Filtering Prompt

In Figure 5, we demonstrate the best-performing prompt in our filtering process over the distant supervised subset of DocRED.

**Reasoning Prompt - Distant Supervision Filtering**

To determine whether the main sentence contains information about the given relation, both the main sentence and the context will be provided. The goal is to identify whether there is evidence of the relation in the main sentence, supported by the context. If there is no relation or the evidence exists solely in the context without requiring the main sentence, respond with No. Otherwise, respond with Yes. Provide reasoning to support your response.

**Context**:
**Main Sentence**: James Michael Osting ( born April 7 , 1977 ) is a former Major League Baseball pitcher .
**Relation**: ("Osting", "date of birth", "April 7 , 1977")
**Evidence**: The relation indicates that Osting was born on April 7, 1977. The main sentence explicitly mentions that Osting was born on April 7, 1977. The answer is Yes.

**Context**: Splashdown is a Hot Tuna album released in 1984 containing the tracks from a previously unreleased live acoustic performance that had been played on the short - lived radio station WQIV in the mid-1970s . During the recording , news of the Apollo - Soyuz mission returning to Earth after the first USA - USSR rendezvous in space reached the station , and the astronauts ' radio transmissions were played at the same time as Jorma and Jack continued with " Police Dog Blues . " The transmissions mixed with the song were preserved for this release as the last track of side 1 .
**Main Sentence**: The album was Hot Tuna 's first release on Relix Records , and one of the first Relix releases .
**Relation**: ("Hot Tuna", "country of origin", "USA")
**Evidence**: The relation indicates that the origin of Hot Tuna is the country of the United States. There is no evidence in the main sentence regarding the country of origin of Hot Tuna. The answer is No.

**Context**:
**Main Sentence**: The Chemung Canal Bank Building is located at 415 East Water Street in Elmira , Chemung County , New York , United States .
**Relation**: ("Chemung County", "capital", "Elmira")
**Evidence**: The relation indicates that Elmira is the capital of Chemung County. The main sentence only specifies the location of Elmira within Chemung County but does not mention Elmira as the capital of Chemung County. The answer is No.

**Context**: Carrie Lam Cheng Yuet - ngor , GBM , GBS (; born 13 May 1957 ) is the 4th and current Chief Executive of Hong Kong . Before that she was the Chief Secretary for Administration , the most senior rank of principal officials of Hong Kong , from 2012 to 2017 .
**Main Sentence**: After graduating from the University of Hong Kong , Lam joined the civil service in 1980 and served in various bureaux and departments .
**Relation**: ("Lam", "educated at", "University of Hong Kong")
**Evidence**: The relation indicates that Lam received education at the University of Hong Kong. The main sentence mentions that Carrie Lam Cheng Yuet-ngor graduated from the University of Hong Kong. The answer is Yes.

**Context**: Pacific Fair is a major shopping centre in Broadbeach Waters on the Gold Coast , Queensland , Australia . It was Queensland 's largest regional shopping centre until 2006 . Pacific Fair was developed by Hooker Retail Developments and opened in 1977 on what was swampland with 96 specialty stores and two anchor tenants . Since then , Pacific Fair has undergone numerous expansions and has grown to have more than 300 specialty stores and four anchor tenants . In January 2014 , work began on a major redevelopment project to meet the predicted regional growth on the Gold Coast . Prior to the redevelopment , the shopping centre had four main major stores including a four - level Myer , Kmart , Target , Coles and Toys ' R ' Us . Daimaru operated in the centre before its Australian withdrawal , albeit briefly .
**Main Sentence**: It also had a 12-screen Birch Carroll and Coyle Cinema ( re - opened as Event Cinemas in late 2015 ) .
**Relation**: ("Event Cinemas", "country", "Australia")
**Evidence**: The relation indicates that Event Cinemas is located in the country of Australia. The main sentence mentions that Event Cinemas is part of Pacific Fair which is located in Australia. The answer is Yes.

**Context**: Benjamin Winslow Harris ( November 10 , 1823 - February 7 , 1907 ) was a nineteenth - century politician , lawyer and judge from Massachusetts . He was the father of Robert Orr Harris . Born in East Bridgewater , Massachusetts , Harris pursued an academic course at Phillips Academy , Andover , graduating in 1847 . He graduated from Dane Law School of Harvard University in 1849 . He was admitted to the bar in Boston , Massachusetts in 1850 , commencing practice in East Bridgewater . He served in the Massachusetts Senate in 1857 , was a member of the Massachusetts House of Representatives in 1858 , was district attorney for the southeastern district of Massachusetts from 1858 to 1866 and was collector of internal revenue for the second district of Massachusetts from 1866 to 1873 . Harris was elected a Republican to the United States House of Representatives in 1872 , serving from 1873 to 1883 , not being a candidate for renomination in 1882 . There , he served as chairman of the Committee on Naval Affairs from 1881 to 1883 . Afterwards , he resumed practicing law in East Bridgewater , Massachusetts and was judge of probate for Plymouth County , Massachusetts from 1887 to 1906 .
**Main Sentence**: Harris died in East Bridgewater on February 7 , 1907 and was interred in Central Cemetery in East Bridgewater .
**Relation**: ("Benjamin Winslow Harris", "place of birth", "East Bridgewater")
**Evidence**: The relation indicates that Benjamin Winslow Harris was born in East Bridgewater. The main sentence lacks information about Benjamin Winslow Harris's place of birth. The evidence for East Bridgewater as his birthplace is exclusively found in the context, not in the main sentence. The answer is No.

**Context**: Greatest Hats is the first compilation album by the Canadian new wave / synthpop group Men Without Hats , released in 1996 .
**Main Sentence**: A slightly modified version of the album was released in the US in 1996 , entitled Collection .
**Relation**: ("Collection", "performer", "Men Without Hats")
**Evidence**: The relation indicates that Men Without Hats is the performer of the Collection album. The main sentence says that Men Without Hats released slightly modified version of the Greatest Hats album which is the album Collection. The answer is Yes.

**Context**: Aaron Hobart ( June 26 , 1787 - September 19 , 1858 ) was a U.S. Representative from Massachusetts . Born in Abington , Massachusetts , Hobart pursued classical studies and graduated from Brown University in 1805 . He studied law , was admitted to the bar and commenced practice in Abington . He served as member of the Massachusetts House of Representatives and served in the Massachusetts State Senate . Hobart was elected as a Democratic - Republican to the Sixteenth Congress to fill the vacancy caused by the resignation of Zabdiel Sampson . He was reelected as a Democratic - Republican to the Seventeenth Congress , elected as an Adams - Clay Republican to the Eighteenth Congress , and reelected as an Adams candidate to the Nineteenth Congress , and served from November 24 , 1820 , to March 3 , 1827 . He declined to be a candidate for renomination in 1826 .
**Main Sentence**: He then served as an Executive councilor 1827 - 1831 and served as probate judge 1843 - 1858 .
**Relation**: ("Aaron Hobart", "date of death", "1858")
**Evidence**: The relation indicates that Aaron Hobart passed away in the year 1858. The main sentence does not contain information about the given relation. The evidence of Aaron Hobart's date of death in 1858 is solely present in the context and is not mentioned in the provided main sentence. The answer is No.

**Context**: [[context]]
**Main Sentence**: [[focus_sentence]]
**Relation**: ("[[entity1]]", "[[relation]]", "[[entity2]]")
**Evidence**:

Figure 5: The prompt for the distant supervision dataset filtering. This prompt includes the natural representation of the relation, the reasoning, and the final answer.

