# OpenReview forum: "MemLLM: Finetuning LLMs to Use Explicit Read-Write Memory"
_TMLR — Accepted by TMLR_

### Review · Reviewer_EYDD · 2025-01-23

**Summary Of Contributions:**

The paper introduces MemLLM, a novel method of enhancing LLMs by integrating a structured and explicit readand-write memory module. MemLLM tackles the aforementioned challenges by enabling
dynamic interaction with the memory and improving the LLM’s capabilities in using stored
knowledge. Experiments indicate that MemLLM enhances the LLM’s performance and
interpretability, in language modeling in general and knowledge-intensive tasks in particular.

**Audience:**

Yes

**Claims And Evidence:**

Yes

**Requested Changes:**

Please see the weakness part.

**Strengths And Weaknesses:**

Strength:
1. MemLLM has both read and write access to the memory component, i.e., it can store information in the
memory as it processes text (or interacts with a user) and retrieve it when it needs it.

2. It specifies an API for read and write access. The LLM issues read commands to retrieve from the
memory and write commands to write to the memory.

3. Its memory has an explicit structured schema, similar to a database schema: interpretable， editable，scalable，and interoperable

Weakness:

1. The method part is very hard to read. For example, the notations are undefined in the paragraph of Memory writes and read. The figures are also hard too read. I suugest the authors to reorganize this part to improve the clearance.

2. Inefficient experiments: (1) For LLMs, the paper only considers Mistral-7B. Larger LLMs are needed. (2) Compare to other KE methods, there is a  performance gap in reliability. While the authors provided some analysis. I think some. discussion on improving it is needed.

---

> ### Author Response · Authors · 2025-02-14
>
> [Undefined notations]: Could you pinpoint which notation is undefined so we could clarify and also apply edits?
>
> [Regarding larger LLMs]: The 7B scale is a popular model size that we can train within our academic resource constraints. As the training data will be publicly available, those with additional compute resources can finetune larger models independently and share their insights. Moreover, we hypothesize that smaller models may benefit more from our approach, while larger models can see diminishing returns [1].
> - [1]: Li, Jiatao, et al. "Evaluating Self-Generated Documents for Enhancing Retrieval-Augmented Generation with Large Language Models." arXiv preprint arXiv:2410.13192 (2024).
>
> [Other analysis]: In Section 4.2, we analyzed the impact of each design factor on MemLLM and discussed its implications for memory redundancy. For further improvements, we could discuss potential future directions, such as the benefits of dynamic relation systems compared to the current static approach. These discussions can be highlighted as future work.

---

### Review · Reviewer_pBKB · 2025-02-02

**Summary Of Contributions:**

This work aims to tackle the limitation of existing language models (whose parametric knowledge can be incomplete and outdated) by using the concept of structured external memory, which can store facts in their triplet formats, e.g., (head entity, relation, tail entity). The major contribution that this work makes over existing works is to train the language model to perform read and write operations over the external memory, and, more specifically, during the write operation, the trained language models extract the facts over the sentences, meanwhile, for the read operation, the stored facts within the external memory are retrieved based on their embedding-level similarities with the query and augmented to the language models. The authors validate the proposed method, called MemLLM, on the perplexity-based evaluation task and the knowledge editing task, showing that it outperforms its variants (or its ablated versions).

**Audience:**

Yes

**Broader Impact Concerns:**

While the authors do not present the Broader Impact Statement, there is no specific concern associated with the proposed approach.

**Claims And Evidence:**

No

**Requested Changes:**

Please see the Weaknesses above.

**Strengths And Weaknesses:**

### Strengths
* The idea to finetune the language model to perform read and write operations over the external memory through the natural language expression is sound.
* The illustrations in Figure 2 clearly show how to perform memory read and write operations.
* The ablation of the proposed approach in Table 2 is extensive, which clearly shows what is the advantage of the proposed approach and its room for improvements.

---

### Weaknesses
* The approach to extracting the vectors for triplet elements is not clear. More specifically, how to store the vectors for the extracted entities? If the entities appear over multiple sentences and documents, their vector representations differ across them. And, how to handle this case?
* It is not clearly discussed whether there is an advantage to building and utilizing the external memory (consisting of the triplet-form facts) over the approach that directly retrieves and uses the raw documents (without converting them into the external memory). Experimental validations on it may be needed.
* The discussion about the related work [1] is not so clear. Why this related work is task-specific (unlike this work), and what are the major contributions that this work makes over this related work?
* The proposed approach to storing the facts in the external memory is similar to the approach that extracts triplets over documents for knowledge base construction [2, 3]. Also, similarity, the proposed approach to using the facts stored in the external memory can be viewed as augmenting the language model with the knowledge retrieved from the external knowledge base (or graph) [4]. However, this work lacks discussions about them and, furthermore, the proposed approach may be potentially compared against them.
* The experimental validation is weak. Specifically, the baseline chosen is one (except for the naive language model prompting baseline without external knowledge), while there are many studies on (parametric or non-parametric) memory-augmented language model approaches, which are also discussed in the Related Work section. Also, for the knowledge editing task, while the authors claim the weaknesses of the existing approaches for it, the authors do not provide experimental evidence on it (i.e., the proposed approach can then handle the limitation of existing approaches in handling multiple facts updated). Lastly, it may be beneficial to evaluate the proposed approach with other language models (other than only one used in this work, namely Mistral-7B)
* The process to generate the memory-read and memory-write data discussed in Section 3.3 is a bit complex. Explaining them with Algorithms may be beneficial.
* When storing the element in the memory, how to handle the entity with the same name but different meanings? For example, the word Apple can indicate the fruit or the company.

---

[1] ChatDB: Augmenting LLMs with Databases as Their Symbolic Memory.

[2] Extract, Define, Canonicalize: An LLM-based Framework for Knowledge Graph Construction.

[3] LLMs for Knowledge Graph Construction and Reasoning: Recent Capabilities and Future Opportunities.

[4] Knowledge-Augmented Language Model Prompting for Zero-Shot Knowledge Graph Question Answering.

---

> ### Author Response · Authors · 2025-02-14
>
> [Vector extraction]: When the memory-write function extracts triplets from a given text, the entities and relation type of that triplet are embedded independent of the context. For instance, in Figure 2(a) when the triplet (Alla Mia Età>>performer>>Tiziano Ferro) is extracted first as mentioned in 3.1, we check if the entities are already existent in the entities table. We do the same thing for the relationship as well. If an  entity (e.g. Alla Mia Età) is new, it will be stored in the entities table with its vector representation extracted via **Contriever** embedding model (mentioned in 3.1). (Same thing for relationship.)
>
> [Advantage to building and utilizing the external memory]: (1) Memory redundancy reduction benefits (cf. last paragraph 4.2) (2) Knowledge editing: in a RAG setup, to edit a single fact, you need to locate all forms of the fact in all stored raw texts--which itself is an extensive task-- and then also update and re-index the embeddings of all affected texts which is computationally intensive.
>
> [On related work #1 task specificness]: In [1], ChatDB gets a database structure as an input. So already we should know the task specific DB schema  which it requires to be defined and prompted to the model. In MemLLM, the memory structure works much more generalized. It is capable of handling language modeling  and also QA tasks (as mentioned in the knowledge editing section) without any alterations to the approach and the structure of the memory. Also the very nature of prompt-based approaches (against finetuning)  make it less faithful and dependent on the content of the memory.
>
> [Comparison to 2,3,4]: The main advantage of MemLLM is that it’s an end-to-end approach that contains both construction and reasoning using triples in language modeling task. It is similar to what [3] proposes as future direction in AutoKG. One can use the training data to finetune a model without any architectural modifications.
>
> [Experimental validation]: To ensure a proper and fair comparison with other methods, we evaluate MemLLM’s effectiveness in multi-fact knowledge editing. In this evaluation, other knowledge editing methods that we mentioned also have undergone the same amount of edits (multi-fact updating) but they show lower performance than MemLLM (Table 3).
> That said, we would like to understand what kind of evaluation the reviewer believes would be more appropriate—while also ensuring fairness in comparison to our approach. Most memory-based methods (or methods such as RAGs and RALMs) rely on pre-populated KBs rather than extracting facts directly via an LLM. Additionally, their memory and compute efficiency are not directly comparable (see Section 4.2 on memory redundancy reduction benefits). Moreover, applying edits to existing facts in an unstructured KB further requires an extensive search process, along with customized, example-based modifications—an approach that can be highly resource-intensive.
>
> [Entity with the same name but different meanings]: Apple as a fruit won’t be considered as an entity, therefore neither in memory-writes nor -reads would be used as an entity if it doesn't have an entity role in the context. But also as mentioned in 3.3 -- Memory write data, we use full mentions in training. This means the model learns to extract full mentions. Apart from that, relation types could also help distinguishing between triplets.

---

> ### Comment · Reviewer_pBKB · 2025-03-03
>
> Thank you for your response. I have carefully read it and have some follow-up comments.
>
> * In the vector extraction part, if I understand correctly, the proposed approach extracts the triplet and then embeds its entities and relation independently (without considering its context). That is, for the two triplets: (Rock, composed_of, Minerals) and (Rock, genre_of, Music), the entity "Rock" has the same embedding despite it having multiple meanings, which may be a clear limitation of the proposed memory-based approach.
>
> * I still believe the current set of baselines is weak, as also pointed out by Reviewer KCAv. I list some potential baselines that may be discussed and compared [1-4], and the response from the authors is not clear enough: the authors mention that their method is an end-to-end approach that contains both construction and reasoning using triples in language modeling tasks, and I am still wondering what is the (motivational and empirical) advantage of the proposed approach over prior works. Also, in the next response, while the authors claim that their approach extracts facts via LLMs unlike others that use pre-populated KBs, I don't see clear benefits of using the extracted facts (that may be noisy and whose process itself is costly) in contrast to the case where we use pre-populated KBs (that are off-the-shelf and available widely), at least in the current set of benchmark experiments. I hope that, given that there are a lot of studies on memory-based language modeling, the authors may further clarify why those are not included in the main experiment (Table 2): the current main table has the ablated variants of the proposed approach without memory.

---

> > ### Author Response · Authors · 2025-03-18
> >
> > [On Vector Representation]: While MemLLM does not model contextual word meanings in the entities, it retrieves facts using both the entity and the relation. Since the query itself is derived from the context, this naturally constrains the search space, ensuring that different relations (e.g., composed_of vs. genre_of) lead to distinct fact retrievals.
> >
> > [On Baselines]: We thank the reviewer for their willingness to engage with
> > us! Below we discuss in detail why we do not think that
> > [1-4] are appropriate baselines for our approach.
> >
> > [1] ChatDB: Augmenting LLMs with Databases as Their Symbolic
> > Memory. (Hu et al., 2023)
> >
> > In ChatDB, the user defines a database schema and a prompt
> > corresponding to the schema for a specific task to be
> > addressed.  The user has to do this from scratch for each
> > new task.  In contrast, MemLLM is designed for general
> > language modeling, making it adaptable to a variety of tasks
> > without extensive prompt engineering.  This means that we
> > can solve tasks such as question answering without changing
> > the memory structure and without designing new prompts.
> >
> > [2] Extract, Define, Canonicalize: An LLM-based Framework
> > for Knowledge Graph Construction. (Zhang & Soh, 2024)
> >
> > [4] Knowledge-Augmented Language Model Prompting for
> > Zero-Shot Knowledge Graph Question Answering. (Baek et al,
> > 2023)
> >
> > The knowledge graphs in [2] and [4] are a structured format
> > that can be interpreted as a memory of triples, similar to
> > our memory. However, while these papers exemplify work on
> > extracting triples from text into knowledge graphs using
> > LLMs [2] and on LLMs “interactively” querying knowledge graphs
> > to answer questions [4], our approach is distinguished by
> > teaching the LLM memory-write and memory-read functionality
> > through finetuning. This makes memory an integral component
> > of the model’s text processing, while enabling the model at
> > the same time to flexibly handle new and updated knowledge.
> >
> > [3] LLMs for Knowledge Graph Construction and Reasoning:
> > Recent Capabilities and Future Opportunities. (Zhu et al.,
> > 2024)
> >
> > This survey from the perspective of the knowledge graph
> > community discusses (among other capabilities and
> > opportunities) systems that are similar to our approach in
> > that the integration of interactive reading and writing
> > capabilities is considered -- specifically this is the case
> > for the proposed AutoKG architecture.  However, to the best
> > of our knowledge our system is the first that goes beyond a
> > conceptual proposal and demonstrates empirical success in
> > both language modeling and knowledge editing.
> >
> > It is important to point out that -- in addition to these
> > features that differentiate us from the knowledge graph
> > literature -- our published training dataset can be used to
> > endow any trainable language model with explicit memory
> > without requiring architectural changes.
> >
> > [Comments on Table 2 and comparison to prior work]:
> > The purpose of Table 2 is to ablate the major components of
> > MemLLM. For example, it quantifies how errors in the
> > extraction of triples affect the overall system.
> >
> > The main evaluation of MemLLM in an application is **Table 3**
> > for the task of knowledge editing. As far as we can see the
> > prior work on memory-based language modeling is not suitable
> > as a baseline here. We are relying on a key property of
> > MemLLM: the fact that its memory is explicit.
> >
> > We would appreciate if the reviewer could comment on how we
> > could compare MemLLM to prior memory work on this
> > task. Please note that Table 3 does compare to prior work
> > relevant to this task.

---

### Review · Reviewer_KCAv · 2025-02-03

**Summary Of Contributions:**

This paper introduces  a novel method for enhancing LLM by integrating an explicit read-write memory module, called MemLLM. The key contributions of the paper include: 1)  interpretable, editable, and scalable memory module for LLMs; 2) improves language modeling performance, particularly in knowledge-intensive tasks (lower perplexity); 3) better knowledge editing performance. MemLLM addresses several limitations of current LLMs, including difficulties in memorizing rare events, updating memory over time, and preventing hallucination.

**Audience:**

Yes

**Claims And Evidence:**

Yes

**Requested Changes:**

I agree with authors that "In a real-world scenario, the size of the memory and, consequently, the size of query results will get large." and Table 2 shows a relatively small negative effect of memory size. If an figure specifically for controlling variable analysis on the impact of scaling memory size could be added, it would more clearly illustrate the trend brought about by further scaling.

Authors make a thorough related work discussion. But the baselines used in the experiments are simple and not enough, could you provide more baseline method results or explain why some previous methods can not be adopted for comparison?

**Strengths And Weaknesses:**

## Strengths
1. The explicit and structured memory module (and its read and write method) is an innovation, providing a clear and interpretable way to store and retrieve knowledge.
2. The experiments are sound and the results are promising -- MemLLM demonstrates substantial improvements in language modeling tasks and knowledge editing tasks.
3. The memory's scalability and interoperability make it a versatile solution that can be adapted to various tasks and LLMs.

## Weaknesses
1. Not enough baseline methods for fair comparison (especially for knowledge editing tasks).
The current implementation of MemLLM handles only 96 relation types, which may limit its applicability to more complex knowledge extraction tasks.
2. As mentioned in the Limitation section, the model does not support composite relations inferred from multiple extracted relations.

---

> ### Author Response · Authors · 2025-02-14
>
> [On limited relation types & other baselines]: The MemLLM data augmentation method is applicable to any DocRE dataset. The current limitation of 96 relation types stems from the seed dataset (DocRED/Re-DocRED). Expanding to an RE dataset with a wider range of relations would enable MemLLM to cover more relation types. For instance, one can use domain specific DocRE datasets to finetune a domain specific model for special applications.
> Despite this limitation, we demonstrate MemLLM's effectiveness by outperforming multiple knowledge editing methods in a multi-editing setup without limiting the test dataset.
>
> [Scaling effect figure]: Thanks for the suggestion! We shall add this plot to illustrate how the trend would be in further scaling.

---

### Author Response · Authors · 2025-02-23

Dear Reviewers and Action Editor,

A revised version has been uploaded with the following updates (highlighted in blue in the paper):
- Algorithm 1 in Appendix C now details the process of generating MemLLM’s memory-read training data, as described in Section 3.3. (In response to reviewer pBKB’s suggestion and reviewer EYDD’s comment on the method's readability.)
- Added a figure on memory scaling efficiency gains, as recommended by reviewer KCAv.
- Added some details on related work, addressing reviewer pBKB’s feedback.

Thank you for your valuable insights and suggestions.

---

> ### Author Response · Authors · 2025-03-18
>
> Dear Reviewers and Action Editor,
>
> A new revision has been uploaded, now including additional related work ([1]–[4], as noted by reviewer pBKB) that provides a broader comparison of MemLLM with other approaches.
>
> Once again, we appreciate the reviewers' valuable insights and suggestions.

---

> > ### Comment · Action_Editor_Gjgw · 2025-04-12
> > **Small error**
> >
> > "While methods like ChatDB Hu et al. (2023)"
> > Incorrect use of citet instead of citep in limitations.
> >
> > Also, as a soft suggestion, the part of the intro quoted below is kind of disjointed. The paragraph breaks are a bit hard to follow and the mention of read/write access followed by the discussion of the API doesn't flow well. Please consider revising.
> >
> > > In this paper, we introduce MemLLM, an LLM endowed with an explicit memory component. It has the general advantages of some of the memory-focused work we discussed: we can keep information accessible indefinitely, beyond the context window, including infrequent information that standard LLMs struggle with.
> > >
> > > The LLM has both read and write access to the memory component, i.e., it can store information in the memory as it processes text (or interacts with a user) and retrieve it when it needs it.
> > >
> > > We specify an API for read and write access. The LLM issues read commands to retrieve from the memory and write commands to write to the memory"

---

> > > ### Author Response · Authors · 2025-04-16
> > >
> > > Thank you for pointing out the issue.
> > >
> > > We have also made a slight revision to the introduction, as suggested, to improve fluency.

---

### Decision · Action_Editor_Gjgw · 2025-03-17

**Recommendation:** Accept with minor revision

**Comment:**

This paper introduces a read-write memory to LLMs. A model can perform write operations after seeing relations expressed in text, then subsequently read from that memory while processing later text, enabling it to surface relevant entities for use in next-token prediction.

Reviewers generally liked the proposed method of the paper. I think the paper presents a clear idea and is well-written. The main issue with the paper is raised by reviewers KCAv and pKBK, who point out that there are other baselines or competing methods that could be compared to.  Reviewer pBKB points out some closely related prior work. In particular, many practitioners may choose to use an approach like RAG over text, or a more explicitly pipelined text -> KB -> RAG over the KB system. The restriction to the 96-relation inventory in DocRED/Re-DocRED seems like a major limitation that would cause practitioners not to adopt this approach, as other labeled datasets for other relation inventories may not be readily available. Some of these weaknesses are discussed in the paper, but not in a centralized way, which seems to me that it doesn't give the whole picture here.

**Requested change:  Please add discussion of how the approach conceptually overlaps with ChatDB and the other papers mentioned by pBKB in the Limitations section. I would also suggest moving content from footnote 4 into the Limitations section and being more up front about the strengths and weaknesses of RAG.**

(Comparison to these baselines would also be great, but I don't think this is strictly required given the current framing of the paper.)

Please also upload a clean copy without any blue formatting; thanks!

**Audience:**

Yes, this is a topic within scope for TMLR and of interest to its audience

**Claims And Evidence:**

Yes, I believe the claims are supported. The paper claims

"Our experiments indicate that MemLLM enhances the LLM’s performance and interpretability, in language modeling in general and knowledge-intensive tasks in particular."

The introduction makes similar claims and does not specify very particular baselines or alternative approaches. It seems to me that broader claims of the superiority or generality of this approach would not be supported, but the paper stops short of claiming this.